# The Effects of Creatine Supplementation Combined with Resistance Training on Regional Measures of Muscle Hypertrophy: A Systematic Review with Meta-Analysis

**DOI:** 10.3390/nu15092116

**Published:** 2023-04-28

**Authors:** Ryan Burke, Alec Piñero, Max Coleman, Adam Mohan, Max Sapuppo, Francesca Augustin, Alan A. Aragon, Darren G. Candow, Scott C. Forbes, Paul Swinton, Brad J. Schoenfeld

**Affiliations:** 1Department of Exercise Science and Recreation, CUNY Lehman College, Bronx, NY 10468, USA; rburke3320@gmail.com (R.B.); alec.pinero@gmail.com (A.P.); colemanmax888@gmail.com (M.C.); adam.mohan@lc.cuny.edu (A.M.); maxsapuppo46@gmail.com (M.S.); francesca.augustin@lc.cuny.edu (F.A.); 2Department of Family and Consumer Sciences, California State University, Northridge, CA 91330, USA; alaneats@gmail.com; 3Faculty of Kinesiology and Health Studies, University of Regina, Regina, SK S4S 0A2, Canada; darren.candow@uregina.ca; 4Department of Physical Education Studies, Faculty of Education, Brandon University, Brandon, MB R7A 6A9, Canada; forbess@brandonu.ca; 5School of Health Sciences, Robert Gordon University, Aberdeen AB10 7AQ, UK; p.swinton@rgu.ac.uk

**Keywords:** muscle thickness, muscle cross-sectional area, strength training, nutritional supplements, lean mass

## Abstract

The purpose of this paper was to carry out a systematic review with a meta-analysis of randomized controlled trials that examined the combined effects of resistance training (RT) and creatine supplementation on regional changes in muscle mass, with direct imaging measures of hypertrophy. Moreover, we performed regression analyses to determine the potential influence of covariates. We included trials that had a duration of at least 6 weeks and examined the combined effects of creatine supplementation and RT on site-specific direct measures of hypertrophy (magnetic resonance imaging (MRI), computed tomography (CT), or ultrasound) in healthy adults. A total of 44 outcomes were analyzed across 10 studies that met the inclusion criteria. A univariate analysis of all the standardized outcomes showed a pooled mean estimate of 0.11 (95% Credible Interval (CrI): −0.02 to 0.25), providing evidence for a very small effect favoring creatine supplementation when combined with RT compared to RT and a placebo. Multivariate analyses found similar small benefits for the combination of creatine supplementation and RT on changes in the upper and lower body muscle thickness (0.10–0.16 cm). Analyses of the moderating effects indicated a small superior benefit for creatine supplementation in younger compared to older adults (0.17 (95%CrI: −0.09 to 0.45)). In conclusion, the results suggest that creatine supplementation combined with RT promotes a small increase in the direct measures of skeletal muscle hypertrophy in both the upper and lower body.

## 1. Introduction

Creatine (methylguanidine-acetic acid) is considered to be one of the few efficacious ergogenic dietary supplements for augmenting resistance training (RT) adaptations [1]. Mechanistically, creatine supplementation increases the skeletal muscles’ total creatine (free creatine and phosphocreatine), allowing for a greater capacity to rapidly resynthesize adenosine triphosphate and consequently enhance high-intensity exercise [1]. Furthermore, creatine influences insulin-like growth factor-1, myogenic regulatory factors, satellite cells, cellular hydration, calcium and protein kinetics, glycogen content, inflammation, and oxidative stress [2], which may contribute to muscle accretion over time [3]. Longitudinal evidence indicates that creatine supplementation, in conjunction with RT, augments gains in muscular strength [4,5], power output [6], and performance in a variety of physical tests related to anaerobic metabolism [7].

Several meta-analyses have investigated the combined effects of creatine supplementation and RT, operationally defined as “a form of physical activity that is designed to improve muscular fitness by exercising a muscle or a muscle group against external resistance” [8], on changes in whole-body lean mass, as assessed by methods such as dual-energy X-ray absorptiometry (DXA), hydrodensitometry, whole-body air displacement plethysmography, and bioelectrical impedance analyses [2,9,10,11]. Collectively, a combination of creatine supplementation and RT results in greater gains in lean mass compared to RT and a placebo. However, lean mass is an imprecise proxy surrogate for skeletal muscle mass as it comprises all non-fat tissue, including body water. Indeed, DXA, often considered to be a gold-standard measure of lean mass [12], correlates relatively poorly with longitudinal hypertrophic changes, as assessed by site-specific imaging modalities [13,14], which are regarded as gold-standard measures for assessing muscle size [15].

Research indicates that creatine supplementation increases total body water [16]. Given that creatine acts as an osmolyte, it is generally believed that the majority of its hydrating effects are compartmentalized intracellularly [17]. However, some evidence suggests that at least some of the lean mass gains from creatine supplementation can be attributed to water retention, perhaps mediated by a decreased urine output [18]. For example, supplementation with 20 g/day of creatine for 3 days followed by 5 g/day for 7 days in untrained participants increased the DXA estimates of their lean mass [17]. More recently, Bone et al. [19] found that supplementation with 20 g/day of creatine for 5 days followed by 3 g/day altered muscle metabolites and water content, which influenced the estimates of lean mass during a time period where minimal changes in the muscle protein mass were likely. Accordingly, the use of lean mass as a proxy for hypertrophy may be particularly problematic in studies investigating the effects of creatine supplementation when combined with regimented RT. To address this issue, we carried out a systematic review and meta-analysis of the current literature on the combined effects of RT and creatine supplementation on the regional changes in muscle mass in studies that utilize direct imaging measures of hypertrophy. Moreover, we performed regression analyses to determine the potential influence of covariates.

## 2. Methods

### 2.1. Literature Search

We preregistered our methods for both the systematic review and the meta-analysis on the Open Science Framework website (https://osf.io/c7bez and https://osf.io/hdqzb (accessed on 15 January 2023), respectively). We searched the PubMed/MEDLINE, Scopus, and Web of Science databases from inception to February 2023 to locate relevant studies. The search syntax was performed using the following combination of terms: (“creatine”) AND (“supplement*”) AND (“resistance training” OR “resistance exercise” OR “weight lifting” OR “weight-lifting” OR “weightlifting” OR “strength exercise” OR “strength training” OR “strengthening” OR “resistive exercise” OR “resistive training”) AND (“muscle hypertrophy” OR “muscular hypertrophy” OR “muscle mass” OR “muscle size” OR “muscle thickness” OR “cross-sectional area” OR “cross sectional area” OR “muscle volume”). The other articles included in our systematic review were either known by the authors or identified by manually searching the bibliographies of the retrieved articles. Two researchers (RB and AM) screened the retrieved abstracts and reviewed the full texts for studies that conceivably met the inclusion criteria. Inclusion required agreement between both researchers; in cases where a disagreement arose, a third researcher (BJS) resolved the dispute. The methods followed the guidelines set forth by the Preferred Reporting Items for Systematic Reviews and Meta-Analyses (PRISMA) [20]. 

### 2.2. Inclusion/Exclusion Criteria 

We included studies that: (1) investigated the longitudinal effects of creatine supplementation (in any form) combined with RT vs. RT without creatine supplementation; (2) had a duration of ≥6 weeks; (3) included adults (≥18 years of age); (4) were published in English-language peer-reviewed journals; and (5) reported pre–post study changes in site-specific hypertrophy, employing a validated imaging modality including magnetic resonance imaging (MRI), computed tomography (CT), or ultrasound. 

Studies were excluded if: (1) the participants had pre-existing musculoskeletal disorders, cardiovascular diseases, or any other condition that could be considered detrimental to the RT performance; (2) other potentially anabolic ingredients were included in the supplementation formula (since protein is a food source, we allowed for supplementation with this macronutrient, provided its provision was equated between conditions); (3) blood flow restriction was incorporated into the RT protocol; or (4) there was insufficient numerical or graphical data provided to assess the differences between the conditions. 

### 2.3. Data Coding and Analysis

We extracted the data from the included studies and coded them in an Excel spreadsheet (Microsoft Corporation, Redmond, Washington), which was performed by 2 authors (FA and MC) using the following classifications: (1) study characteristics (author, year of publication, and sample size); (2) participant demographics (age, sex, and RT status); (3) training methods (sets, exercises, frequency, duration, and repetitions); (4) supplementation methods (dose, timing, blinding, placebo, and protein supplementation); and (5) pre- and post-training means and standard deviations of hypertrophy. In cases where the studies lacked sufficient information regarding pre–post changes, we contacted the authors to request the missing data. If we were unable to acquire this data from authors, we extracted values from the figures using the WebPlotDigitizer online software (https://apps.automeris.io/wpd/, (accessed on 15 March 2023)) where applicable. To account for the possibility of coder drift, a third researcher (MS) re-coded 30% of the studies, all of which were randomly selected for assessment [21]. The per case agreement was determined by dividing the number of variables that were coded the same by the total number of variables. Acceptance required a mean agreement of ≥90%. Any discrepancies in the extracted data were resolved through discussion and the mutual consensus of the coders.

### 2.4. Methodological Quality

As previously described [22], we assessed the methodological quality of the included studies via the Downs and Black assessment tool [23], which is a 27-item checklist that addresses the following aspects of a study’s design: reporting (items 1–10), external validity (items 11–13), internal validity (items 14–26), and statistical power (item 27). Consistent with previous systematic reviews of exercise interventions, we modified the checklist by adding two items relating to participant adherence (item 28) and training supervision (item 29) [24,25,26]. Each item in the checklist was scored with a “1” if the criterion was satisfied or with “0” if the criterion was not satisfied. Based on the summary scores, the studies were classified as follows: “good quality” (21–29 points); “moderate quality” (11–20 points); or “poor quality” (less than 11 points) [25,26]. Three reviewers (AP, FA, and AM) independently rated each study; any disagreements in the study ratings were resolved by a majority consensus.

### 2.5. Statistics

A Bayesian framework was chosen over a frequentist approach as it can provide more flexible modeling, enabling the results to be presented intuitively through the reporting of subjective probabilities [27]. Where sufficient data were available, the comparative effects comparing RT with and without creatine supplementation were quantified using controlled absolute mean differences effect sizes to facilitate interpretations. Where there was a need to combine the results on different scales, controlled standardized mean differences effect sizes were used. Three-level random-effect Bayesian hierarchical models were used to pool the effect sizes and model the average effects, variance within the studies, variance between the studies, and covariance of multiple outcomes reported in the same study (e.g., hypertrophy of different sites and/or outcomes reported at multiple time points following the baseline). Within-study variance is influenced by pre–post correlations [28] that are generally not reported. Rather than specifying a single correlation value, this was estimated but constrained using informative prior distributions. We intended to use these informative prior distributions for the comparative effect sizes based on previous meta-analysis data [29]. However, due to limited information regarding regional changes in muscle mass, we ultimately employed default weakly informative prior distributions.

The inconsistency in the models was described by comparing the variances across the three levels. Inferences from all the analyses were made from posterior samples generated using the Hamiltonian Markov Chain Monte Carlo method and via the use of credible intervals (CrI), and the probabilities were calculated. The interpretations were based on the range of values within the CrI and the calculations of the probability that the magnitude of the average effect size exceeded the qualitative thresholds (i.e., small, medium, and large) that were specific to the strength and conditioning interventions [29]. Meta-regression or subgroup analyses were performed when there were sufficient data, including a minimum of 4 data points per category level or 10 data points for the continuous variables [30]. The small-study effects (publication bias, etc.) were visually inspected with funnel plots and quantified with a multi-level extension of Egger’s regression intercept test [31]. The analyses were performed using the R wrapper package *brms* interfaced with *Stan* to perform the sampling [32]. 

## 3. Results

### 3.1. Descriptive Data

A total of 10 articles met the inclusion criteria (see Figure 1 and Table 1). The duration of the studies ranged from 6 to 52 weeks. Four studies included young adults (aged 21–26 years) [33,34,35,36] and 6 studies included older adults (aged 57–72 years) [37,38,39,40]. Four studies included only males [35,37,38,39], one study included only females [40], and five studies included both males and females [33,34,36,41,42]. Two studies employed resistance-trained participants [34,36] and the others employed untrained participants. All the studies incorporated a parallel group design and all the RT sessions were performed two–five times per week. One study solely focused on training the elbow flexors [35]; all the other studies implemented total-body training protocols. One study involved a creatine “loading” phase, which involved the consumption of 20 g/day of creatine for 5 consecutive days prior to consuming 5 g/day for the remaining duration of the study [35]; one study involved the supplementation of either 6 g/day of creatine or 6 g/day of creatine in combination with 30 g of whey protein [41]; and all the other studies implemented dosing protocols of either 0.1 or 0.15 g/kg/day of creatine. Three studies involved the ingestion of creatine two–four times per week [33,36,39] and all the other studies involved the ingestion of creatine five–seven times per week. One study measured the muscle CSA of the lower leg and forearm using peripheral quantitative computed tomography [42], one study measured the CSA of the vastus lateralis using ultrasonography [41], and the remaining studies measured the muscle thickness of the upper and lower extremities using ultrasonography.

### 3.2. Univariate Analysis of Standardized Differences

An analysis of the controlled standardized mean differences was conducted by pooling all the outcomes and measurement scales. A total of 44 outcomes were included across the 10 studies meeting the inclusion criteria (elbow flexors: nine; elbow extensors: eight; knee flexors: eight; knee extensors: eight; ankle plantarflexors: four; ankle dorsiflexors: three; vastus lateralis: two; forearm: one; and lower leg: one). A forest plot of the meta-analysis is presented in Figure 2, with a pooled mean estimate of 0.11 (95%CrI: −0.02 to 0.25) providing evidence for a very small effect favoring creatine supplementation (*P*(>0) = 0.961, *P*(>0.1) = 0.588, *P*(>0.2) = 0.089). Substantial heterogeneity was identified (τ = 0.10 (75%CrI: 0.03 to 0.20)), with central estimates indicating a low covariance between multiple outcomes reported from the same study (ICC = 0.17 (75%CrI: 0.01 to 0.79)). Egger’s regression intercept test produced wide intervals and a visual inspection of the funnel plot (Figure 3) did not identify any small-study-related issues.

These distributions represent “shrunken estimates” based on all the effect sizes included and the random effects model fitted and borrowed information across the studies to reduce uncertainty. The black circles and connected intervals represent the median values and 95% credible intervals for the shrunken estimates. The white circles and intervals represent the raw estimates and sampling variances calculated directly from the study data.

### 3.3. Multivariate Analysis of Absolute Differences

The initial analyses were conducted using the controlled absolute changes in the muscle hypertrophy measured by an ultrasound in cm. The multivariable analyses were conducted and split into upper-body (elbow extensors: seven studies [33,34,36,37,38,39,40]; elbow flexors: eight studies [33,34,35,36,37,38,39,40]) and lower-body (knee extensors and knee flexors: seven studies [33,34,36,37,38,39,40]).

The meta-analysis results for the upper body are illustrated in Figure 4, providing evidence favoring creatine supplementation. The marginal pooled controlled mean difference estimates were 0.16 cm (95%CrI: −0.10 to 0.39) and 0.10 cm (95%CrI: −0.13 to 0.32) for the elbow extensors and flexors, respectively. Substantive heterogeneity was identified (elbow extensors: τ = 0.15 cm (75%CrI: 0.04 to 0.31); elbow flexors: τ = 0.13 cm (75%CrI: 0.04 to 0.27)) with a positive but uncertain correlation (ρ = 0.38 (75%CrI: −0.11 to 0.84)). The probability that the controlled absolute mean differences favored creatine supplementation for both the extensors and flexors was $P\left(>0\right)=$ 0.801.

The meta-analysis results for the lower body are illustrated in Figure 5 and also provide evidence favoring creatine supplementation. The marginal pooled controlled mean difference estimates were 0.13 cm (95%CrI: −0.07 to 0.37) and 0.11 cm (95%CrI: −0.06 to 0.31) for the knee extensors and flexors, respectively. Substantive heterogeneity was identified (knee extensors: τ = 0.11 cm (75%CrI: 0.03 to 0.24); knee flexors: τ = 0.07 cm (75%CrI: 0.02 to 0.17)) with a positive but uncertain correlation (ρ = 0.18 (75%CrI: −0.32 to 0.66)). The probability that the controlled absolute mean difference favored creatine supplementation for both the extensors and flexors was $P\left(>0\right)=$ 0.850.

### 3.4. Analysis of Moderating Effects

The potential moderating effects of age and the duration of the intervention were investigated using controlled standardized mean differences across all the outcomes. Six studies comprising 26 outcomes were conducted with older adults (mean age: 61.6 years) and four studies comprising 18 outcomes were conducted with younger participants (mean age: 23.5 years). Evidence was obtained that the controlled standardized mean differences were greater for younger participants ($\beta_{Younger}=$ −0.17 (95%CrI: −0.45 to 0.09); *P*(Younger > Older) = 0.910; Figure 6).

Seven studies comprising 30 outcomes were conducted with short interventions (mean duration: 8.6 weeks, range: 6–16 weeks) and three studies comprising 14 outcomes were conducted with long interventions (52 weeks). A substantive overlap was obtained for the controlled standardized mean difference distributions, with limited evidence of greater values for a short intervention ($\beta_{Short:Long}=$ −0.09 (95%CrI: −0.37 to 0.19); *P*(Short > Long) = 0.767; Figure 6).

### 3.5. Study Quality

A qualitative assessment of the studies via the Downs and Black checklist indicated a median score of 20 (range: 17 to 25 points). Four studies were deemed to be of good quality [35,37,38,40], six studies were classified as being of moderate quality [33,34,36,39,41,42], and no studies were found to be of poor quality.

## 4. Discussion

This is the first meta-analysis to examine the regional changes in muscle accretion from a combination of creatine supplementation and RT. The pooled analysis of the supplementation and resistance protocols included in this review indicated that creatine supplementation enhances regional skeletal muscle hypertrophy when combined with structured RT. However, compared to a placebo, the magnitude of this effect was trivial to small (ES = 0.11), with a fairly narrow 95% CrI (−0.02, 0.25); as such, the practical significance and implications of this on an individual level are likely small. Previous meta-analyses performed on creatine supplementation and RT have shown significant increases in whole-body lean mass over time (1.1 to 1.4 kg) [2,9,10,11], with larger standardized mean differences (SMD range: 0.24–0.42). These larger SMDs may be associated with creatine’s effect on increasing total body water; as such, it is possible that a portion of the observed increases in the lean mass may reflect the accumulation of extracellular fluid, as opposed to muscle hypertrophy. Alternatively, considering that lean mass measures take into account all the non-fat tissues of the entire body, it is also conceivable that hypertrophy occurred in regions not assessed by the direct imaging modalities or in other tissues (e.g., bone [42]). These hypotheses warrant further investigation.

To determine the potential influence of covariates on our findings, we carried out subanalyses of body region, age, and study duration. In regard to the hypertrophy of individual muscle groups, our subanalyses revealed that creatine supplementation has similar effects on the upper and lower body musculature, as well as on the limb flexors and extensors, irrespective of the body region. This contrasts with DXA-derived evidence showing that creatine supplementation augmented RT gains in the upper body lean mass compared to the lower body (upper: 7.1% vs. lower: 3.2%) in resistance-trained men [43]. Syrotuik and Bell [44] found that “responders” to creatine supplementation had a higher percentage of type II muscle fibers (responders averaged 63.1% and non-responders averaged 39.5%), thus one could speculate that specific muscles that possess a higher percentage of type II muscle fibers may have a greater hypertrophic response. However, our results suggest that any specific differences between the muscles of an individual with regard to their fiber type do not appreciably alter the effectiveness of creatine. This is consistent with evidence that the majority of the body’s musculature has relatively similar fiber type percentages [45]. Moreover, the magnitude of the effects of individual muscle groups were similarly small, with median improvements relative to the placebo ranging from 0.10 to 0.16 cms, which would suggest that the discrepancies between the whole body lean mass changes observed in previous meta-analyses compared to the present findings, using direct measures of site-specific hypertrophy, are unlikely to be due to the regions not assessed. However, future research is warranted to verify these speculations and assess other muscles (e.g., torso musculature). 

A subanalysis evaluating the influence of age revealed a modest benefit for creatine supplementation combined with RT in young adults (ES = 0.23 (95%CrI: 0.01, 0.44)) compared to older adults (ES = 0.06 (95%CrI: −0.10, 0.21)). Syrotuik and Bell [44] found that responders to creatine supplementation had a greater proportion of type II muscle fibers, as well as lower baseline levels of muscle creatine content. Evidence indicates that age-related remodeling of motor units is primarily due to the denervation of type II muscle fibers, which may, in theory and supported by our data, reduce the responsiveness to creatine supplementation. However, our findings are in contrast with a previous meta-analysis that observed similar gains between younger and older adults in whole-body measures of their lean masses (young: 1.2; older: 1.1 kg) [9]. Future research is warranted to compare the site-specific responses between younger and older adults. Overall, the practical implications of our findings that showed greater gains in young adults is questionable given the relatively modest magnitude of the effect between these populations. 

A subanalysis of study duration found a greater benefit in shorter (≤10 weeks) compared to longer (≥16 weeks) supplementation protocols. However, the differences between these timeframes were trivial (median values of 0.15 and 0.06 for shorter and longer durations, respectively) and likely of little practical significance. Moreover, this finding is confounded by the fact that all the studies with longer time frames included older participants; thus, any effect, if one does indeed exist, may therefore be attributed to age as opposed to study duration. 

Our meta-analysis had several limitations that must be acknowledged. First, only one study investigated the hypertrophic effects of creatine supplementation in women only. Evidence indicates that men respond more favorably to supplementation than women for increases in lean mass [9]. Whether these sex-related differences are also specific to muscle hypertrophy remains to be determined. Second, only two studies involved resistance-trained individuals. Conceivably, those with RT experience may be able to train harder and thus derive a greater benefit from creatine supplementation. This hypothesis warrants further investigation. Third, there is considerable interindividual variability in response to creatine supplementation, with increases in muscle creatine concentrations ranging from 2 to 40 mmol/kg of dry mass [46]. Greenhaff et al. [47] reported that approximately 20 to 30% of subjects are “nonresponders”, which is defined by an intramuscular creatine content that increases by less than 10 mmol/kg of dry mass after a loading phase. The characteristics that may underpin this nonresponse to creatine supplementation include high initial muscle creatine levels, a low percentage of type II fibers, a low muscle fiber CSA, and a low fat-free mass [44]. Therefore, the modest effects of creatine supplementation on skeletal muscle hypertrophy may be at least partially due to a lack of delineation between responders and nonresponders within the studies included in the present analysis. Future longitudinal trials that correlate the changes in hypertrophy and intramuscular creatine content warrant more research. 

## 5. Conclusions

A pooled analysis of the current data suggests that creatine supplementation promotes a small increase in skeletal muscle hypertrophy in both the upper and lower body musculature when combined with a regimented RT program. Those considering creatine supplementation for the goal of regional muscle hypertrophy should consider the practical significance of the small magnitude of effect. Furthermore, young adults appear to derive a greater hypertrophic benefit compared to older individuals, but the magnitude of this difference is relatively modest, calling into question the practical relevance of this finding. Future longitudinal studies should endeavor to assess the combined effects of creatine supplementation and RT on intra- vs. extracellular fluid accumulation and intramuscular creatine content, with respect to site-specific direct imaging measures of hypertrophy in both young and older adults and include both male and female participants.

## Figures and Tables

**Figure 1 nutrients-15-02116-f001:**
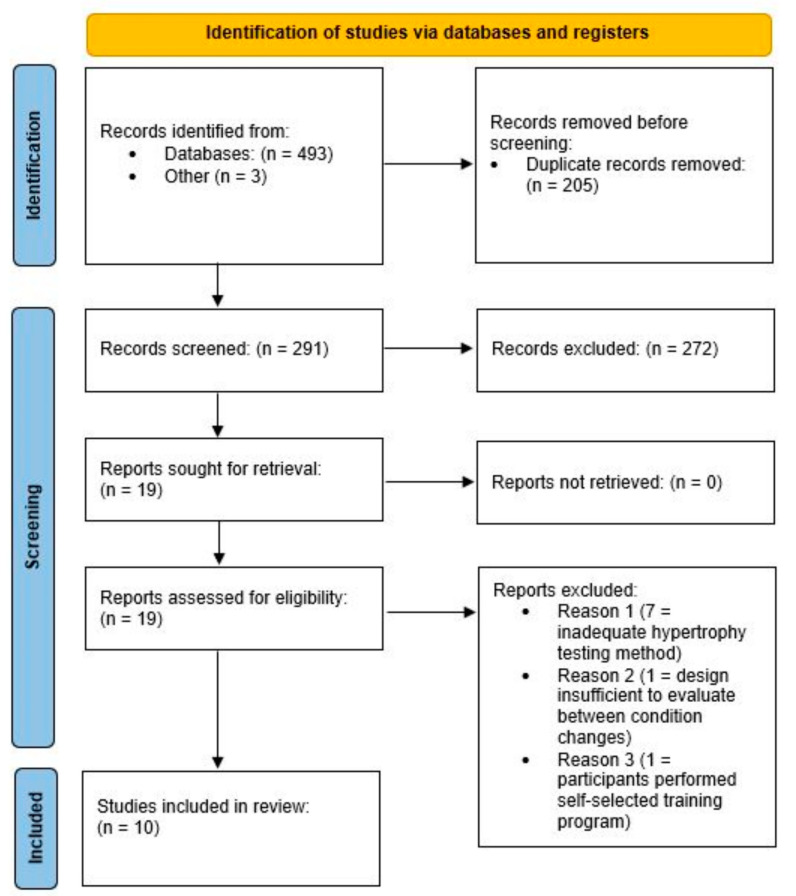
PRISMA flow chart of the search process.

**Figure 2 nutrients-15-02116-f002:**
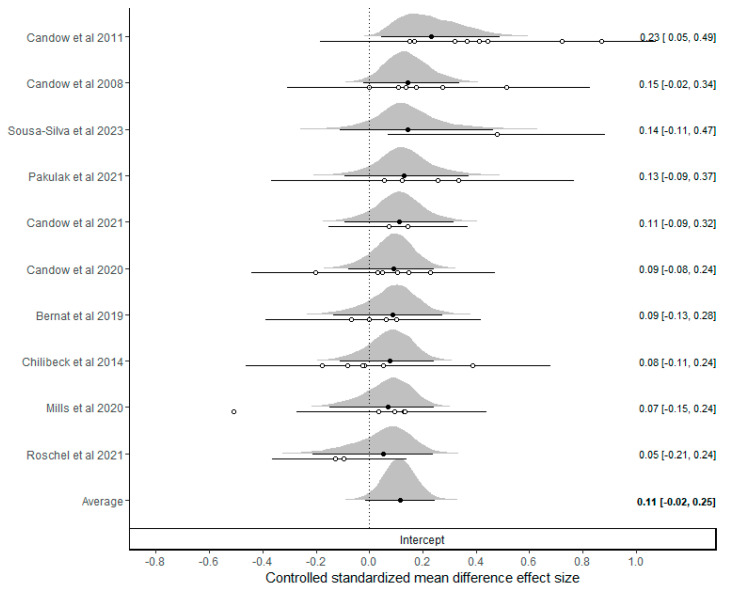
Bayesian forest plot of controlled standardized mean difference effect sizes across all outcomes [33,34,35,36,37,38,39,40,41,42].

**Figure 3 nutrients-15-02116-f003:**
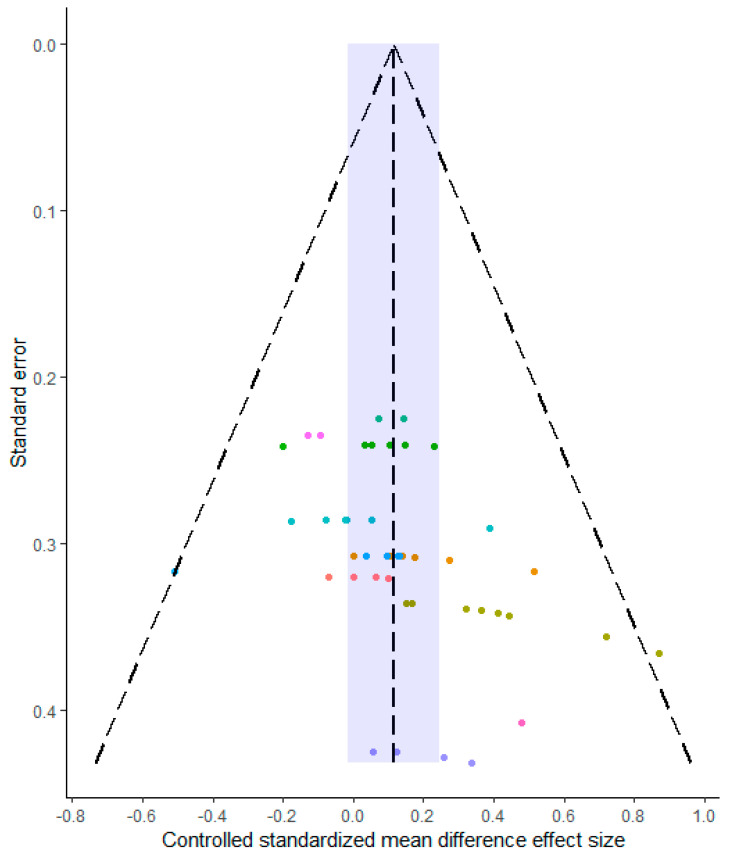
Funnel plot of all standardized effect sizes. Data are colored according to the individual studies. Blue region illustrates the pooled mean estimate and 95% credible interval.

**Figure 4 nutrients-15-02116-f004:**
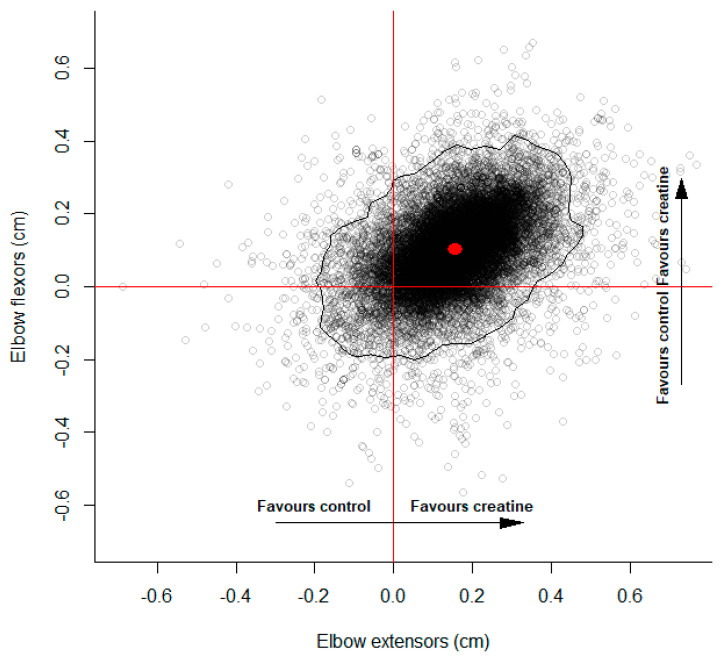
Multivariate controlled mean difference posterior estimates for elbow flexors and extensors. Points represent multivariate mean difference posterior estimates. Positive values favor creatine supplementation and negative values favor control. Red dot illustrates median values for the marginal mean difference posterior estimates. Black curve represents the 95% highest posterior density region.

**Figure 5 nutrients-15-02116-f005:**
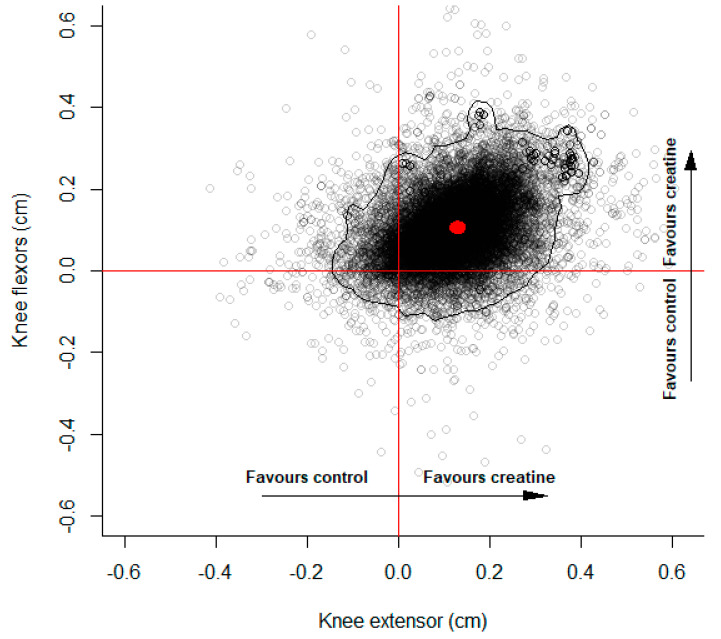
Multivariate controlled mean difference posterior estimates for knee flexors and extensors. Points represent multivariate mean difference posterior estimates. Positive values favor creatine supplementation and negative values favor control. Red dot illustrates median values for the marginal mean difference posterior estimates. Black curve represents the 95% highest posterior density region.

**Figure 6 nutrients-15-02116-f006:**
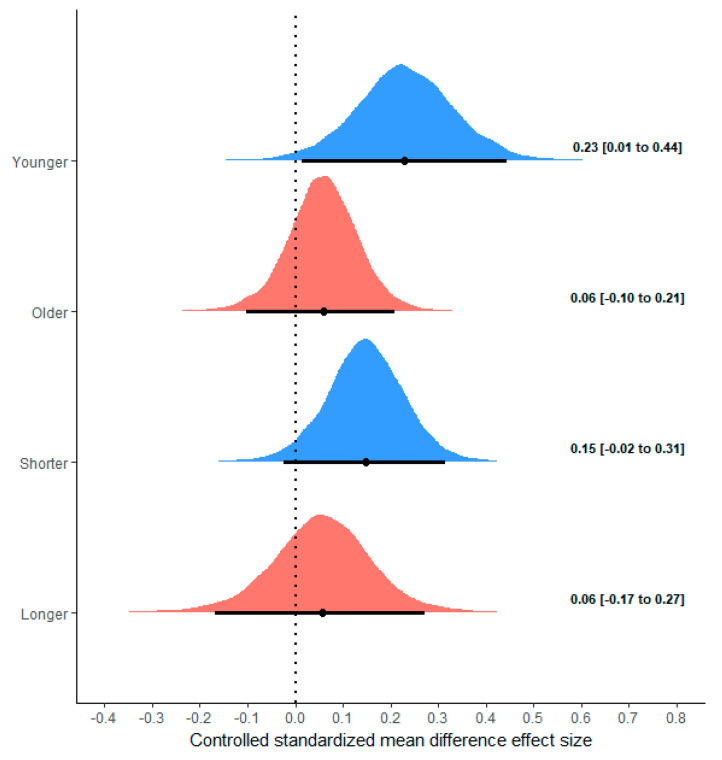
Distributions of controlled mean difference effect sizes according to age of participants and duration of study. Distributions represent posterior estimates of pooled mean difference effect size. Results illustrate distributions obtained from two separate meta-regression models, the first with a group variable for age, and the second with a group variable for duration.

**Table 1 nutrients-15-02116-t001:** Summary of the methods and results of included studies.

Study	Sample	Design	RT Protocol	CR Protocol	Duration	Results
Bernat et al. [37]	24 older, untrained men	Random assignment to 1 of 2 groups: (1) CR + RT; (2) PLA + RT	Total body unsupervised protocol performed 2 d/wk consisting of 3–4 sets per exercise at 80% 1 RM with 2 min inter-set rest intervals	0.1 g/kg/d consumed post-training and at participants’ leisure on non-training days	8 wks	− Similar between-group changes in muscle thickness of the elbow flexors, elbow extensors, knee flexors, and knee extensors
Candow et al. [33]	38 young, physically active, untrained men and women	Random assignment to 1 of 4 groups: (1) CR + RT performed twice/wk; (2) CR + RT performed thrice/wk; (3) PLA + RT performed twice/wk; (4) PLA + RT performed thrice/wk	Total body protocol performed 2–3 d/wk consisting of 2–3 sets per exercise of 10 repetitions with 1–2 min inter-set rest intervals	0.15 g/kg/d for participants in 2 d/wk RT and 0.10 g/kg for participants in 3 d/wk RT; no mention of supplementation on non-training days	6 wks	− Superior increases in elbow flexor muscle thickness in the CR + RT groups (16–20%) compared to PLA + RT groups (2–6%). − Superior increases in elbow extensor muscle thickness in the CR + RT groups (26–27%) compared to PLA + RT groups (11–13%) − Similar increases in knee flexor muscle thickness in CR + RT groups (10–17%) and PLA + RT groups (6–15%) − Superior increases in knee extensor muscle thickness in CR + RT groups (11–12%) compared to PLA + RT groups (3–5%)
Candow, Chilibeck, Gordon, Vogt et al. [38]	46 older, untrained men	Random assignment to 1 of 2 groups: (1) CR + RT; (2) PLA + RT	Total body supervised protocol performed 3 d/wk consisting of 3 sets of 10 repetitions per exercise at 80% 1 RM	0.1 g/kg/d consumed in 2 equal doses pre- and post-training and with food on non-training days	12 months	− Similar between-group changes in muscle thickness of the elbow flexors, elbow extensors, knee flexors, knee extensors, ankle plantarflexors, and ankle dorsiflexors
Candow et al. [39]	25 older, untrained men	Random assignment to 1 of 2 groups: (1) CR + RT; (2) PLA + RT	Total body supervised protocol performed 3 d/wk consisting of 3 sets of 10 repetitions with 2 min inter-set rest intervals	0.1 g/kg/d consumed in 3 equal doses pre- and post-training and before bed on training days; no supplementation on non-training days	10 wks	− Superior increases in elbow extensor muscle thickness in the CR + RT group (11.6%) compared to the PLA + RT group (1.4%) − Superior increases in knee flexor muscle thickness in the CR + RT group (9.4%) compared to the PLA + RT group (3.2%) − Superior increases in knee extensor muscle thickness in the CR + RT group (11.3%) compared to the PLA + RT group (5.8%) − Superior increases in ankle plantarflexor muscle thickness in the CR + RT group (13.8%) compared to the PLA + RT group (8%) − Similar between-group changes in muscle thickness of the elbow flexors and ankle dorsiflexors
Candow, Chilibeck, Gordon & Kontulainen [42]	52 older, untrained men and women	Random assignment to 1 of 2 groups: (1) CR + RT; (2) PLA + RT	Total body supervised protocol performed 3 d/wk consisting of 3 sets of 10 repetitions per exercise	0.1 g/kg/d consumed in 2 equal doses pre- and post-training and with food on non-training days	12 months	− Similar between-group changes in muscle CSA of the forearm and lower leg
Chilibeck et al. [40]	33 postmenopausal, untrained women	Random assignment to 1 of 2 groups: (1) CR + RT; (2) PLA + RT	Total body supervised protocol performed 3 d/wk consisting of 3 sets of 10 repetitions per exercise	0.1 g/kg/d consumed in 2 equal doses pre- and post-training and with food on non-training days	12 months	− Similar between-group changes in muscle thickness of the elbow flexors, elbow extensors, knee flexors, knee extensors, ankle plantarflexors, and ankle dorsiflexors
Mills et al. [34]	22 young, recreationally trained men and women	Random assignment to 1 of 2 groups: (1) CR + RT; (2) PLA + RT	Total body protocol performed 5 d/wk consisting of 3 sets per exercise of 6–10 repetitions with 2 min inter-set rest intervals	0.1 g/kg/d consumed during training sessions (5 d/wk); no supplementation on non-training days	6 wks	− Similar between-group changes in muscle thickness of the biceps brachii, triceps brachii, quadriceps femoris, hamstrings, and gastrocnemius
Pakulak et al. [36]	13 young, resistance-trained men and women	Random assignment to 1 of 2 groups: (1) CR + RT; (2) PLA + RT	Total body protocol performed 4 d/wk consisting of 3 sets per exercise of 6–10 repetitions with 2 min inter-set rest intervals	0.1 g/kg/d consumed 60 min prior to training sessions (4 d/wk); no supplementation on non-training days	6 wks	− Superior increases in knee extensor muscle thickness in the CR + RT groups (10.8%) compared to PLA + RT groups (5.7%) − Similar between-group changes in muscle thickness of the elbow flexors, elbow extensors, and knee flexors
Roschel et al. [41]	88 elderly, pre-frail and frail men and women	Random assignment to 1 of 4 groups: (1) CR + RT; (2) PLA + RT; (3) CR + Whey + RT; (4) Whey + RT	Total body supervised protocol performed 2 d/wk consisting of 2–4 sets at 50–70% 1 RM	6 g/d in CR + RT and 6 g/d + 30 g Whey in CR + Whey + RT; supplements consumed 7 d/wk	16 wks	− Similar between-group changes in muscle CSA of the vastus lateralis
Sousa-Silva et al. [35]	17 young, untrained men	Random assignment to 1 of 2 groups: (1) CR + RT; (2) PLA + RT	Unilateral elbow flexion protocol (2 d/wk): One arm of each participant performed the biceps curl combined with blood flow restriction for 4 sets of 15–30 reps at 30% 1 RM with 30 s inter-set rest intervals and the other arm performed the biceps curl without blood flow restriction for 4 sets of 10–12 repetitions at 70% 1 RM with 2 min inter-set rest intervals	20 g/d consumed for the first 5 days followed by 5 g/d consumed post-training and during regular training time on non-training days	8 wks	− Superior increases in elbow flexor muscle thickness in CR + RT (17%) compared to PLA + RT (9%)

Abbreviations: CR: creatine; PLA: placebo; and RT: resistance training.

## Data Availability

The data presented in this study are available on reasonable request from the corresponding author.

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
