# Peer review of "The Effects of Creatine Supplementation Combined with Resistance Training on Regional Measures of Muscle Hypertrophy: A Systematic Review with Meta-Analysis"

_nutrients, 2023, doi:10.3390/nu15092116_

Round 1

Reviewer 1 Report

This is a well conducted study and well written paper in which I can detect no obvious problems or needs for revision 

Author Response

Thank you

Reviewer 2 Report

Well written and thought out. I have no suggestions for improvements.

Author Response

Thank you

Reviewer 3 Report

Effects of creatine supplementation combined with resistance training on regional measures of muscle hypertrophy: A systematic review with meta-analysis

Abstract.

This section identifies the subject of the study very well, identifies the steps followed for its verification, summarises the main results obtained and finally states the main conclusion of the research. Excellent work

Lines 22 and 23. Although the journal template allows it, we know it is a spelling mistake to truncate syllables at the end of a line. Please proofread this and all lines of the entire article and avoid these errors that only hurt the writers of the article: you.   Thank you in advance.

Introduction

Lines 34-44. Excellent exposition of Creatine and its effects.

Lines 45-70. Good explanation and references regarding the effects of Creatine supply at different doses and times.

However, the study has an important methodological shortcoming due to its ambiguity in the use of the term "Resistance Training" (RT). This type of training has many expressions that can range from explosive strength with self-loads or very light loads (multisprings, plyometrics, multi-throwing, reaction with small sprints...) or maximum strength requirements (powerlifting, weightlifting, maximum lifts in bodybuilding...) Not having taken this into account in their research is an important bias.

Your treatment of the object of study is as if you were doing a systematic review and meta-analysis of aerobic exercise in the loss of fat mass... Which aerobic exercise: swimming, running, skating, walking, rowing... for how long, always at the same intensity or varied within each session... ? This ambiguity does not belong in science because it does not allow the study problem to be properly identified.

Methods 

Literature search

Ok, excellent explanation and excellent search strategy.

Inclusion/exclusion criteria

Lines 92-97. In relation to my final comment in the "Introduction" section, "type of strength training" is missing as an essential criterion. If I had included it, it would have changed the whole article and they could have arrived at " real certains".

Data coding and analysis

Ok, excellent explanation and excellent strategy

Methodological quality

Ok, excellent explanation and excellent strategy

Statistics

Ok, excellent explanation and excellent strategy

Results

Descriptive data

The 10 articles finally selected are very heterogeneous and creatine absorption capacity is not the same at 21 years of age compared to 72 (the factors that determine creatine uptake and retention within the cell are not only the level of saturation of intramuscular deposits before starting supplementation, but also other aspects linked both to the subject and to the way it is ingested: 1. - Age, intramuscular creatine deposits tend to decrease with advancing age, in fact middle-aged subjects (< 40 years) tend to respond better to creatine supplementation than young people (20 to 30 years) (Mesa, et al. 2001).

2. - When the supplement is taken in relation to the hours of training and rest, as the benefits are really significant when the supplement is taken immediately before, during or at the end of the training session, whereas the effect tends to be lost as the intake moves away from the training hours (Cribb & Hayes 2006)) as well as when the difference between the differences between the daily doses of Creatine supply and its relation to the initial muscle mass (The amount of total muscle mass of the subjects, as subjects with greater muscle development tend to have higher proportions of fast fibres or in general may respond with greater sensitivity to creatine supplementation (American College of Sport Medicine 2000, Syrotuik & Bell 2004) is not taken into account. ) 

In table 1 the studies of:  Candow et al (2011); Candow et al (2008); Candow, Chilibeck, Gordon & Kontulainen (2021); Chilibeck et al (2015); Mills et al (2020); Pakulak et al (2021), that is to say 60% of the works you have selected do not identify what type of strength training they have used and the other 40% worked with intensities ranging from 30% 1RM to 80% 1RM which implies very, very, very different stimuli to the muscle and its potential hypertrophy which is what you are ultimately looking to compare. 

Discussion and Conclusion

Being based on a search that needs improvement, the findings declared by the authors respond to a reality very different from the one they really wanted to evaluate due to the lack of rigour in the search, so they cannot be said to be " really true".

Author Response

Reviewer #3:

Effects of creatine supplementation combined with resistance training on regional measures of muscle hypertrophy: A systematic review with meta-analysis

Response: Thank you very much for your thorough and insightful review. We have addressed your comments on a point-by-point basis, hopefully to your satisfaction. 

Abstract.

This section identifies the subject of the study very well, identifies the steps followed for its verification, summarises the main results obtained and finally states the main conclusion of the research. Excellent work

Lines 22 and 23. Although the journal template allows it, we know it is a spelling mistake to truncate syllables at the end of a line. Please proofread this and all lines of the entire article and avoid these errors that only hurt the writers of the article: you.   Thank you in advance.

Response: We agree, however this was due to the journal formatting. Our submitted manuscript does not truncate any words at the end of a line, but MDPI creates their own document from our file; unfortunately, we have no control of their formatting.

Introduction

Lines 34-44. Excellent exposition of Creatine and its effects.

Lines 45-70. Good explanation and references regarding the effects of Creatine supply at different doses and times.

However, the study has an important methodological shortcoming due to its ambiguity in the use of the term "Resistance Training" (RT). This type of training has many expressions that can range from explosive strength with self-loads or very light loads (multisprings, plyometrics, multi-throwing, reaction with small sprints...) or maximum strength requirements (powerlifting, weightlifting, maximum lifts in bodybuilding...) Not having taken this into account in their research is an important bias.

Your treatment of the object of study is as if you were doing a systematic review and meta-analysis of aerobic exercise in the loss of fat mass... Which aerobic exercise: swimming, running, skating, walking, rowing... for how long, always at the same intensity or varied within each session... ? This ambiguity does not belong in science because it does not allow the study problem to be properly identified.

Response: Fair point. We have revised to provide the ACSM operational definition of resistance training as follows: “Resistance training is a form of physical activity that is designed to improve muscular fitness by exercising a muscle or a muscle group against external resistance.”

Methods 

Literature search

Ok, excellent explanation and excellent search strategy.

Inclusion/exclusion criteria

Lines 92-97. In relation to my final comment in the "Introduction" section, "type of strength training" is missing as an essential criterion. If I had included it, it would have changed the whole article and they could have arrived at " real certains".

Response: As above, we clarified with an operational definition.

Data coding and analysis

Ok, excellent explanation and excellent strategy

Methodological quality

Ok, excellent explanation and excellent strategy

Statistics

Ok, excellent explanation and excellent strategy

Results

Descriptive data

The 10 articles finally selected are very heterogeneous and creatine absorption capacity is not the same at 21 years of age compared to 72 (the factors that determine creatine uptake and retention within the cell are not only the level of saturation of intramuscular deposits before starting supplementation, but also other aspects linked both to the subject and to the way it is ingested: 1. - Age, intramuscular creatine deposits tend to decrease with advancing age, in fact middle-aged subjects (< 40 years) tend to respond better to creatine supplementation than young people (20 to 30 years) (Mesa, et al. 2001).

Response: Chilibeck et al. (2017) found no differences between young and old with regards to intramuscular creatine stores. Only when they examined vastus lateralis studies only, did older adults have lower intramuscular stores. A recent meta-analysis examining whole body lean tissue mass found similar responses to creatine supplementation in young and old participants (Delpino et al., 2022: DOI: 10.1016/j.nut.2022.111791). We also know, that as participants age they tend to lose fast twitch (type II muscle fibers), which in theory may actually decrease the responsiveness to creatine supplementation as you age and supports our current meta-analysis.

  1. - When the supplement is taken in relation to the hours of training and rest, as the benefits are really significant when the supplement is taken immediately before, during or at the end of the training session, whereas the effect tends to be lost as the intake moves away from the training hours (Cribb & Hayes 2006)) as well as when the difference between the differences between the daily doses of Creatine supply and its relation to the initial muscle mass (The amount of total muscle mass of the subjects, as subjects with greater muscle development tend to have higher proportions of fast fibres or in general may respond with greater sensitivity to creatine supplementation (American College of Sport Medicine 2000, Syrotuik & Bell 2004) is not taken into account. ) 

Response: Timing may play a small role while loading (i.e., creatine ingested in close proximity to training may be superior). However, we would like to note that the Cribb & Hayes (2006) used a multi-ingredient supplement that contained creatine, therefore it still remains to be elucidated whether creatine in close proximity to training is truly superior to ingestion at other times of the day. Overall, based on the current evidence, timing of creatine appears to play a negligible effect (for a recent review see Candow et al., (2022: DOI: 10.3389/fspor.2022.893714), As per the fiber types influencing the fibers, we agree that this may have played a role and help to explain that younger adults may respond more favorably than older adults.

In table 1 the studies of:  Candow et al (2011); Candow et al (2008); Candow, Chilibeck, Gordon & Kontulainen (2021); Chilibeck et al (2015); Mills et al (2020); Pakulak et al (2021), that is to say 60% of the works you have selected do not identify what type of strength training they have used and the other 40% worked with intensities ranging from 30% 1RM to 80% 1RM which implies very, very, very different stimuli to the muscle and its potential hypertrophy which is what you are ultimately looking to compare. 

Response: The compelling body of literature indicates that hypertrophy is achieved across a very broad spectrum of loading ranges that encompass 30% to 80% 1RM (see for example: https://pubmed.ncbi.nlm.nih.gov/28834797/ and https://pubmed.ncbi.nlm.nih.gov/35015560/). Thus, this would not have influenced our findings.

Discussion and Conclusion

Being based on a search that needs improvement, the findings declared by the authors respond to a reality very different from the one they really wanted to evaluate due to the lack of rigour in the search, so they cannot be said to be " really true".

Response: Fair point. We have revised to reflect that our findings are based on the supplementation and resistance protocols included in this review to make clear that we are not trying to extrapolate to other types of training methodologies that may differ in response to creatine supplementation.

Round 2

Reviewer 3 Report

I believe that the authors have answered clearly and provided scientific evidence compared to what was not included in the first version of the paper.

I would therefore like to thank you for having followed my suggestions and for having made a real effort to explain each of the points that I had included in my first review.